# An Improved U-Net Image Segmentation Method and Its Application for Metallic Grain Size Statistics

**DOI:** 10.3390/ma15134417

**Published:** 2022-06-22

**Authors:** Peng Shi, Mengmeng Duan, Lifang Yang, Wei Feng, Lianhong Ding, Liwu Jiang

**Affiliations:** 1National Center for Materials Service Safety, University of Science and Technology Beijing, Beijing 100083, China; pshi@ustb.edu.cn (P.S.); mengduan@xs.ustb.edu.cn (M.D.); yanglifang_xin@163.com (L.Y.); 2Innovation Group of Marine Engineering Materials and Corrosion Control, Southern Marine Science and Engineering Guangdong Laboratory, Zhuhai 519080, China; 3School of Automation, Nanjing University of Science and Technology, Nanjing 211103, China; fw654282696@163.com; 4School of Information, Beijing Wuzi University, Beijing 101149, China; lhdingbwu@sina.com

**Keywords:** complex image, grain size, image segmentation, improved U-Net, metallographic microstructure analysis

## Abstract

Grain size is one of the most important parameters for metallographic microstructure analysis, which can partly determine the material performance. The measurement of grain size is based on accurate image segmentation methods, which include traditional image processing methods and emerging machine-learning-based methods. Unfortunately, traditional image processing methods can hardly segment grains correctly from metallographic images with low contrast and blurry boundaries. Moreover, the proposed machine-learning-based methods need a large dataset to train the model and can hardly deal with the segmentation challenge of complex images with fuzzy boundaries and complex structure. In this paper, an improved U-Net model is proposed to automatically accomplish image segmentation of complex metallographic images with only a small training set. The experiments on metallographic images show the significant advantage of the method, especially for the metallographic images with low contrast, a fuzzy boundary and complex structure. Compared with other deep learning methods, the improved U-Net scored higher in ACC, MIoU, Precision, and F1 indexes, among which ACC was 0.97, MIoU was 0.752, Precision was 0.98, and F1 was 0.96. The grain size was calculated based on the segmentation according to the American Society for Testing Material (ASTM) standards, producing a satisfactory result.

## 1. Introduction

Image segmentation is the process of dividing an image into several regions with similar properties. The traditional segmentation methods mainly include threshold-based segmentation methods [1], region-based segmentation methods [2], edge-detection-based segmentation methods [3], graph-theory-based segmentation methods [4], and energy functional-based segmentation methods [5]. Threshold-based segmentation methods divide image pixels into several categories by setting different feature thresholds, including the histogram threshold segmentation method [6], the iterative determination method [7], the inter-class variance threshold method [8], and the dynamic threshold segmentation method [9]. The region-based segmentation method divides the image into different regions according to the similarity criterion. It mainly includes the watershed segmentation algorithm [10], the level set segmentation algorithm [11], and the clustering segmentation algorithm [12]. The edge-based segmentation method is fast and effective for edge detection, but it is only suitable for images with relatively low noise and is not very suitable for complex data. It mainly includes the Canny operator [13], the Sobel operator [14], the Kirsch operator [15], and the Log operator [16]. The basic principle of the segmentation method based on graph theory is to remove specific edges and divide the graph into several subgraphs to achieve segmentation, mainly including GraphCut [17] and GrabCut [18]. The segmentation method based on energy functional mainly refers to the active contour model [19] and its improved algorithm. The active contour model can obtain a closed and smooth edge curve, but it is difficult to cope with the change of the model’s topology.

With the development of machine learning, some image segmentation based on machine learning arose, which can be subdivided into several directions, including the feature-encoder-based method [20], the region proposal network (RPN [21]), recurrent neural networks (RNNs) [22], upsampling or deconvolution-based method, and DeepLab [23]. RPN is an abbreviation for the region proposal network method, which is a fully convolutional network that simultaneously predicts object bounds and objectness scores at each position. VGGNet [24] and ResNet [25] are two important methods in the field of feature extraction. The core idea of the regional proposal is to detect the color space and similarity matrix and then classify and predict according to the detection results. Ren et al. [26] merged RPN and Fast R-CNN [27] into a single network that achieved state-of-the-art object detection accuracy on PASCAL VOC 2007, 2012, and MS COCO datasets with only 300 proposals per image. RNN [28] is a network with sequence data as the input; recursion in the evolution direction of the sequence and all nodes are connected in a chain. RNN can learn from sequence data over a long time and retain memories along with sequences. This ability makes it useful for many computer vision tasks, including semantic segmentation and data annotation. Le et al. [29] proposed a multiview recurrent neural network (MV-RNN) approach for 3D mesh segmentation. Segmentation methods based on upsampling or deconvolution mainly include FCN [30], U-Net [31], and SegNet [20]. FCN is a fully convolutional network for semantic segmentation proposed by Jonathan Long. It treats images at the pixel level to solve the semantic segmentation problem. U-Net is a semantic segmentation network based on FCN. It can be trained from very few images and outperforms the previous best networks in the ISBI (Cell Image Segmentation by Electron Microscopy) challenge. SegNet is a pixel-level classification network with an encoder network and a corresponding decoder network. Differently from U-Net, SegNet uses pooled indices between the encoder and decoder instead of tensors, which makes SegNet less effective for the segmentation of images with fewer boundary pixels. The Deeplab method combines deep convolutional neural networks (DCNNs [32]) with a probability graph model (DenseCRFs [33]), which can complete per-pixel classification and achieve good image segmentation results. At the same time, there are various other deep learning methods applied to image recognition. DeCost et al. [34] used computer vision methods to collect various microstructural data, develop quantitative microstructural descriptors, and define objective categories of microstructures. Valente et al. [35] proposed a comparative analysis of a multilayer perceptron and self-organizing mapping topology used to segment a microstructure from metallographic images. Huang et al. [36] used the pretrained CNN representation to study the relationship between the processing conditions and the microstructure obtained by descent and visualization techniques. Chen et al. [37] used Mask R-CNN to complete the learning and recognition of potential characteristics of the microstructure of aluminum alloy. The U-Net++ network [38] proposed by Zhou overcomes the two limitations of U-Net: the optimal depth is a priori unknown and imposes unnecessary restrictive fusion schemes. Zhou et al. evaluated U-Net++ using six different medical image segmentation datasets and achieved good results. ResU-Net [39] is a semantic segmentation neural network which combines the strengths of residual learning, and U-Net is proposed for road area extraction. The network is built with residual units and has an architecture similar to that of U-Net. ResU-Net++ [40] is a fully automatic model for pixel-wise polyp segmentation improved from ResU-Net. It is an architecture created to address the need for more accurate segmentation of colorectal polyps found in colonoscopy examinations. A-DenseUNet [41] is an end-to-end image segmentation architecture. It aggregates multiscale semantic information to generate global features and encodes these features together with skip connection features to the decoder side. The model is able to learn multiscale semantic features at each level.

Although the segmentation methods above have been frequently applied to metallographic images, they have not worked well for complex metallographic images. One reason is that machine learning methods need a large number of labeled training images. Another fact is that in complex images such as metallographic images, the background of the segmentation task accounts for a large proportion, but the pixel proportion of the grain boundary that needs to be segmented is very small. Therefore, for such data sets with extremely uneven pixel distribution, it is difficult for the existing loss function to achieve the effect of network optimization, and this poor effect leads to a large deviation in the final measured grain size. Various properties of metals are closely related to the average size of their grains, and a deviation in grain size measurement will cause the predicted metal properties to be seriously inconsistent with the actual metal properties [42]. It is necessary to explore a new approach.

In this paper, an improved U-Net method is proposed for image segmentation, which can deal with complex images with a lower contrast and a blurred boundary. Moreover, the training of the improved U-Net model is simple and only needs a small dataset. Experimental observation shows that the single-phase metal image segmented by the method proposed in this study was much more accurate compared to those segmented by other methods. The potential of this method for quantitative analysis of microstructures was also investigated.

The main contributions of this paper are as follows. First of all, an improved U-Net model trained with a small training dataset was proposed, which worked well for segmentation of complex metallographic images and showed better performance than other models. Secondly, a loss function combining Focal loss and Dice loss was proposed to solve the problem of data imbalance in some datasets like metallographic images. Thirdly, DropBlock [43] and Batch Normalization [44] were used to prevent overfitting, and the Adam optimization algorithm [45] was used to optimize the improved U-Net model. Finally, based on the image segmentation results, the planimetric method and the intercept method were used to calculate the grain size according to the ASTM standard (E112-12), and satisfying results were obtained.

This paper is organized as follows. Section 2 describes the improved U-Net network structure, a specific loss function improvement, the Adam optimization algorithm, and the methods to prevent overfitting. In Section 3, the dataset, selection of evaluation metrics, details of network training, and comparison of experimental results are presented. The application of the method for an automatic rating of grain size also is described. Finally, Section 4 gives the conclusions of our work.

## 2. Improved U-Net

U-Net is a networking and training strategy that uses data augmentation to more efficiently utilize labeled data, which allows it to achieve good results with small sample data. However, the original U-Net model has several shortcomings in the loss function, optimization, and the overfitting problem. An improved U-Net network was proposed, which inherited the advantages of the U-Net model while improving in some other aspects. Firstly, Focal loss and Dice loss were combined as a new loss function of the improved U-Net model. It solved the problem of data imbalance in image segmentation and could achieve good segmentation results with a small training set of only 30 image samples. Secondly, to prevent the problem of overfitting caused by too little data, the DropBlock algorithm and Batch Normalization (BN) algorithm were used. Thirdly, Adam was chosen as the optimization algorithm, which could maximize the performance of the model and achieve the highest score in all evaluation metrics. The structure of the improved U-Net segmentation model is shown in Figure 1.

### 2.1. Improved Loss Function

The original U-Net uses binary cross-entropy [46] as the loss function. Binary cross-entropy is widely used in classification tasks and most semantic segmentation scenarios, but it also has obvious shortcomings. When the pixel distribution of the image is extremely unbalanced, the performance of the cross-entropy is very poor. When the image segmentation task only needs to segment the foreground and the background, the current number of foreground pixels is much smaller than the number of background pixels, that is to say, the number of *y* = 0 is much larger than the number of *y* = 1, which makes the model seriously biased towards the background, leading to poor results.

Lots of metallographic images have a heavy class imbalance (e.g., the boundary pixels only account for 3–5%). This imbalance results in a poor segmentation effect while the accuracy rate is very high. To solve this problem, a new loss function was designed.

Cross entropy examines every pixel in the image and then compares the class prediction (softmax or sigmoid) with the target vector. For dichotomies, the formula is as follows:(1)$$BCE=-\frac{1}{N}\sum_{i=1}^{N}\left(y_{i}\log p_{i}+\left(1-y_{i}\right)\log\left(1-p_{i}\right)\right)$$
where N is the number of samples, $y_{i}$ represents the true sample label, and $p_{i}$ is the predicted probability of the category. Focal loss can solve the problem that the ratio of positive and negative samples is complete unbalanced in image segmentation, whose formula is as follows:(2)$$FL=-\frac{1}{N}\sum_{i=1}^{N}\left(\alpha y_{i}{\left(1-p_{i}\right)}^{\gamma }\log p_{i}+(1-\alpha )\left(1-y_{i}\right)p_{i}^{\gamma }\log\left(1-p_{i}\right)\right)$$
where α is the weight factor, by adding ${\left(1-p_{i}\right)}^{\gamma }$, the loss of the sample with a large prediction probability is reduced, while the loss of the sample with a small prediction probability is increased, so as to strengthen the attention on the positive sample and improve the phenomenon of target imbalance.

The Dice coefficient is derived from Lee Raymond Dice [47]. Dice is a set similarity measurement function, which can be used to calculate the similarity between two samples (value range is [0, 1]). The Dice coefficient can also be viewed as a loss function:(3)$$\mathrm{DC}(A,B)=\frac{2\mathrm{TP}}{2\mathrm{TP}+\mathrm{FP}+\mathrm{FN}}=\frac{2\left|A\cap B\right|}{\left|A\right|+\left|B\right|}$$
where A is the predicted value, and B is the truth value. The meanings of TP, FP, TN, and FN are shown in Table 1.

In this paper, after modifying the Dice coefficient, a suitable Dice loss function can be obtained:(4)$$\mathrm{DL}=\frac{\mathrm{TP}}{\mathrm{TP}+\alpha \mathrm{FN}+\beta \mathrm{FP}}$$

Expand the description of Equation (4), then:(5)$$\mathrm{DL}=\sum_{i=1}^{N}\frac{p_{i}g_{i}}{p_{i}g_{i}+\alpha g_{i}(1-p_{i})+\beta p_{i}(1-g_{i})}$$
where α and β are the trade-offs of penalties for false negatives and false positives, and $g_{i}$ is the ground truth. Combining the Dice loss and Focal loss, then:(6)Loss=λFL+DL

Then expanding the description of this loss function:(7)$$\begin{array}{l} \mathrm{Loss}=\sum_{i=1}^{N}\frac{p_{i}g_{i}}{p_{i}g_{i}+\alpha g_{i}(1-p_{i})+\beta p_{i}(1-g_{i})}-\\ \lambda \frac{1}{N}\sum_{i=1}^{N}\left(\alpha y_{i}{\left(1-p_{i}\right)}^{\gamma }\log p_{i}+(1-\alpha )\left(1-y_{i}\right)p_{i}^{\gamma }\log\left(1-p_{i}\right)\right) \end{array}$$
where λ is the trade-off between Dice loss and Focal loss, and λ is set to be 0.5 based on the performance on the validation set. In some scenarios (such as metallographic images), most negative samples are located in the background area, and few negatives are located in the transition area between the foreground and the background. The samples located in the background area are relatively easy to be classified. Meanwhile, the corresponding score during training is large, and the loss is relatively small. When calculating the reverse gradient, the easy negative example has a limited impact on parameter convergence. As a result, hard negative samples are difficult to be updated and have poor results.

Considering the unbalanced pixels, more weight values are added on the boundary. By combining Focal Loss and Dice Loss and setting the parameters to α=0.25 and β=2, a better effect than that of other loss functions is achieved.

### 2.2. Adam Optimization

The optimization problem is one of the core problems of machine learning. A suitable optimizer can improve efficiency and save computing resources, which can play a key role in improving the speed and effect of machine learning.

In this paper, after comparing different optimizers, Adam was used as the optimizer. Unlike a traditional stochastic gradient descent, which maintains a single learning rate, Adam designs independent adaptive learning rates for different parameters by computing the first- and second-moment estimates of the gradient. It combines the advantages of AdaGrad and RMSProp.

The AdaGrad optimization algorithm is called an adaptive learning rate optimization algorithm. Each time a batch size of data is used for a parameter update, the algorithm calculates the gradients of all the parameters and then accumulates the squared sum of the gradients of the parameters to a variable s with an initial value of 0. When updating this parameter, the learning rate will vary with s.

AdaGrad reserves a learning rate for each parameter to improve performance on sparse gradients, while RMSProp adaptively preserves the learning rate for each parameter based on the mean of the nearest magnitude of the weight gradient.

By combining the advantages of AdaGrad and RMSProp, the Adam optimization algorithm has excellent performance on both nonstationary and online problems. Specifically, the algorithm computes exponential moving averages of gradients, and the hyperparameters beta1 and beta2 control the decay rate of these moving averages. The initial value of the moving average and the beta1, beta2 values are close to 1 (recommended value), so the bias of the moment estimates is close to 0. The bias is boosted by first computing the biased estimate and then computing the bias-corrected estimate.

### 2.3. Prevention of Overfitting 

When the training sample is small while the model is more complex, it often faces the problem of overfitting. Considering that our training data were few, three methods were used to prevent the overfitting problem.

First, more data was invested in training. Data augmentation methods such as panning, scaling, shading, and flipping images were used to augment the data. Specifically, random rotations were adopted in the range (0, 2π) (including horizontal and vertical mirror symmetry) and set the scaling factor between 0.8 and 1.2 for all training and label data. Second, the DropBlock method was used. DropBlock is a regularization method suitable for convolutional layers. The DropBlock method drops features in adjacent and related feature regions that become blocks. It can achieve the purpose of generating a simpler model and introduce the concept of learning partial network weights in each iteration training to compensate for the weight matrix, thereby reducing overfitting. Third, Batch Normalization was used for training. When training with Batch Normalization, you can see that one training example appears with other examples in the mini-batch, and the training network no longer produces deterministic values for a given training example. The same sample and different samples form a mini-batch, and their outputs are different. It means that the output of the same sample is related to both the sample itself and other samples that belong to the same mini-batch as this sample. This correlation alleviates the overfitting problem of the model.

## 3. Results and Discussion

In this study, the improved U-Net was adopted to segment images at a high level of abstraction. The specific work steps are shown in Figure 2. To train a better improved U-Net network, after labeling the image, the given training set was preprocessed, including image size adjustment, Gaussian smoothing, image denoising, contrast enhancement, and data augmentation. After the original image was segmented by the improved U-Net model and other segmentation methods, the segmentation results were compared and analyzed. Firstly, the segmentation effects of different segmentation methods were compared. Secondly, the performance of different loss functions was compared. Thirdly, the grain size of the image segmented by the improved U-Net network was measured, and the measurement error was calculated.

### 3.1. Dataset

In this paper, many typical single grain metallographic images were collected as our experiment dataset, called SGMD. The dataset contained 60 image samples, including 690 alloy, nickel base alloy, low-carbon steel alloy, austenitic steel, pure iron, yellow brass, and ferrite steel phase metallographic pictures.

The dataset was divided into the training set, the validation set, and the test set. The training set contained 30 image samples, the validation set contained 15 image samples, and the test set contained 15 image samples. The dataset was randomly divided one hundred times, and experiments were executed separately. The averaged performance value was used for comparison.

#### 3.1.1. Image Labeling

By using Labelme (https://github.com/wkentaro/labelme, accessed on 11 May 2020), our metallographic images were annotated, and their boundaries were obtained. Figure 3 shows the process of image labeling. First, Labelme was used to label Figure 3a (original image) to generate Figure 3b. Because the red border of the label was too thick, the small crystals were almost covered, so it was necessary to refine the image that was labeled for the first time. After binarization, the skeleton was extracted by the morphological method, and the skeleton was retained as a single pixel, as shown in Figure 3c. After being processed by the skeleton extraction algorithm, the border was only one pixel. The proportion of border pixels was very small in the background, and the training of the model would have been very difficult. The boundaries were thickened with the method of erosion in morphology so that it could be trained normally, and the final result is shown in Figure 3d.

#### 3.1.2. Image Preprocessing

During image preprocessing, median filtering and Gaussian filtering were used for simple denoising. Median filtering is nonlinear filtering that preserves edge information by removing salt and pepper noise, while Gaussian filtering is smooth linear filtering that can effectively remove Gaussian noise.

Since the grains in the metallographic images were very dense and irregular in size, it was difficult to manually mark them, so the data were relatively small. To get good training results with less data, data augmentation was the most effective way. This paper mainly used elastic deformation, translation, rotation, light and dark changes, and other methods to expand the data. With sufficient data in this way, the training effect of the network model could be improved, and the network model had invariance and good robustness. The preprocessed images are shown in Figure 4.

### 3.2. Evaluation Metrics

To accurately measure the performance of segmentation models, several metrics were adopted, including ACC, Precision, Recall, F1-score, the Dice coefficient, and Mean IoU (MIoU). They are widely used in image segmentation, defined as follows.

ACC [48] is the simplest metric, which is the percentage of pixels that are correctly marked relative to the total pixels (Equation (6)).
(8)$$\mathrm{ACC}=\frac{\mathrm{TP}+\mathrm{TN}}{\mathrm{TP}+\mathrm{TN}+\mathrm{FP}+\mathrm{FN}}$$

Precision (also called positive predictive value) is the fraction of relevant instances among the retrieved instances, which shows the correct detection ability of the response model for abnormal data. It is calculated as Equation (7):(9)$$\mathrm{Precision}=\frac{\mathrm{TP}}{\mathrm{TP}+\mathrm{FP}}$$

Recall refers to the ratio of the total number of correctly identified abnormal samples to the number of true abnormal samples. It reflects the ability of the response model to identify abnormal data, calculated by Equation (8):(10)$$\mathrm{Recall}=\frac{\mathrm{TP}}{\mathrm{TP}+\mathrm{FN}}$$

F1-score is a combination of precision and recall. The robustness of the classification model is positively correlated with the F1 score. It is calculated as Equation (9).
(11)$$F1=\frac{2\mathrm{TP}}{2\mathrm{TP}+\mathrm{FP}+\mathrm{FN}}=\frac{2\times \mathrm{Precision}\times \mathrm{Recall}}{\mathrm{Precision}+\mathrm{Recall}}$$

The Dice coefficient is the most commonly used evaluation index in semantic segmentation. The higher the Dice coefficient, the higher the similarity between two samples. The Dice score is defined as Equation (10).
(12)$$\mathrm{DC}(A,B)=\frac{2\mathrm{TP}}{2\mathrm{TP}+\mathrm{FP}+\mathrm{FN}}=\frac{2\left|A\cap B\right|}{\left|A\right|+\left|B\right|}$$
where A is the predicted segmentation and B is the ground truth. Intersection over Union (IoU) is another metric used to evaluate the predictions from an image segmentation model. The IoU is defined as Equation (11).
(13)$$\mathrm{IoU}(A,B)=\frac{\mathrm{TP}}{\mathrm{TP}+\mathrm{FP}+\mathrm{FN}}=\frac{\left|A\cap B\right|}{\left|A\cup B\right|}$$

Mean-IoU is a metric that takes the IoU over all of the classes and takes the mean of them, calculated as Equation (12).
(14)$$\mathrm{MIoU}=\frac{1}{k+1}\sum_{i=0}^{k}\frac{\mathrm{TP}}{\mathrm{FN}+\mathrm{FP}+\mathrm{TP}}$$

### 3.3. Model Training

This part mainly introduces more details regarding the implementation environment, data augmentation strategies, and hyperparameters when training the model.

The model was implemented by the Keras package [49]. In the process of training with the basic U-Net, batch normalization, dropout regularization, weight decay regularization, and data augmentation were combined. To reduce the global brightness difference between different samples and data sets, local histogram equalization was performed on the trained images. The number of channels of each convolution layer was increased by 0.5 times, and the image input dimension was set to 512 × 512 pixels, which improved the fitting and generalization ability of the model.

For data augmentation, the data were randomly rotated in the range (0, 2*π*), including horizontal and vertical mirror symmetry, and scaled between 0.8 and 1.2 for all training data and labeled images to alleviate the overfitting problem.

Correctly tuning the values of hyperparameters is the key to obtaining good performance of machine learning models. According to our experiences, the initial learning rate was set to 0.015, and decay was set according to Equation (13). After much repeated training, the loss no longer fluctuated on both the validation set and the training set after 40 epochs. Consequently, the epoch was set to 50, and batch size was set to 16. The improved training network loss could reach 0.0112, as shown in Figure 5.
(15)$$\mathrm{lr}=\mathrm{lr}\times {\left(\frac{1-\mathrm{iterations}}{\mathrm{total}\_\mathrm{iterations}}\right)}^{0.9}$$

In addition, to achieve good performance on the validation set, adaptive learning rate adjustment and early stopping criteria were also set. The specific rules were as follows.

If there was no improvement in validation loss over five epochs, the learning rate began to decline.

If the validation loss did not improve over the next ten epochs, the training was terminated.

After multiple sets of experimental comparisons, the Focal loss and Dice loss were finally selected as the loss function. When the parameters were set to α = 0.25, β = 2, the best effect could be obtained, and MIoU could reach 0.88. Since the adjustment factor reduces the contribution of simple sample loss, reduces the rate of simple sample weighting, and expands the acceptance range of low loss samples, it is more beneficial for the study of difficult samples.

Table 2 shows the performance of different optimizers. Adam performed the best of all the metrics, so it was chosen as the optimization algorithm.

### 3.4. Performance Comparison with Traditional Image Segmentation Methods

To verify the performance of the proposed method, our method was compared with an advanced material image processing software called Mipar and several representative image segmentation methods, including the Morphological method, the Canny operator, and the Watershed method.

#### 3.4.1. Morphological Method

The Morphological method utilizes morphological operations, such as erosion, dilation, opening, closing, and top-hat transformation, to extract, modify, and manipulate the features presented in the image based on their shapes [50]. In order to facilitate the extraction of grain boundaries and quantitative information, the grayscale image is converted into a binary image. After binarization, there is a lot of point-like salt and pepper noise, which is removed by the median filtering method. In order to preserve the linear boundary, a similar linear grain substructure adopts the dot net division method and uses eight connected neighborhood pixel models to search the image edges.

#### 3.4.2. Canny Operator

Canny operator is a strong edge detection operator proposed by Canny in 1986. It is an optimal approximation operator for the product of SNR and location. Because the Canny operator adopts the hysteresis threshold method, the false edges generated by noise and grain internal texture can be suppressed well, and it has a good effect on the edge detection of the image of the metallographic structure.

#### 3.4.3. Watershed Method

The Watershed algorithm is essentially an image segmentation method based on topological theory. It mainly divides images according to their domain characteristics, but for metallographic images with too much noise, it is easy to produce oversegmentation.

In this metallurgical micrograph image segmentation experiment, the U-Net, U-Net++, and improved U-Net were compared with the traditional segmentation algorithms and typical software (Mapir, San Diego, CA, USA). The specific effect is shown in Figure 6. The result of the Morphological method is shown in Figure 6b. Because the noise was mistaken for time boundary pixels, the boundary of the segmentation result was too thick. The method based on the Canny operator in Figure 6c could clearly detect the boundary, but it was difficult to remove noise, which would eventually affect the final grain size statistics. The Watershed-based image segmentation algorithm in Figure 6d was prone to oversegmentation for images with more noise, so it could not be used in the dataset. Figure 6e is the segmentation result of advanced material image processing software called Mipar (http://keple.cn/170195-170195.html, accessed on 6 June 2021), but the result was very poor. The original U-Net model was very poorly realized in this experiment. As seen from Figure 6f, the segmentation was very fuzzy and the boundary could be recognized basically. The segmentation result of U-Net++ is shown in Figure 6g. Compared with that of U-Net, it had better performance, but it did not make much progress in segmenting complex regions. Figure 6h shows the result of the improved U-Net model, whose segmentation effect was the best, with the noise being completely filtered out. Thus, the improved U-Net was adopted as the segmentation method.

### 3.5. Performance Comparison with Other Deep Learning Methods

In this experiment, our method was compared with several deep learning methods suitable for microscopic image segmentation, including FCN, SegNet, DeepLabV3, Mask R-CNN, U-Net, ResU-Net, A-DenseU-Net, ResU-Net++, and U-Net++. The experimental results are shown in Table 3. Compared with those of other deep learning methods, the improved U-Net achieved good results in all indicators. The improved U-Net scored higher in ACC, MIoU, Precision, and F1 indexes, among which ACC was 0.97, MIoU was 0.752, Precision was 0.98, and F1 was 0.96. The improved U-Net of the proposed loss function could guide the practice to pay more attention to the direction of the boundary pixels. To prevent overfitting, the regularization terms are introduced. In this paper, Adam tuning is chose based on the optimal learning rate and batch size. According to the experiment, Adam tuning have a good performance than other optimizers.

Since the convergence effect is important for a deep learning method, the convergence effects of these networks were compared, and the results are shown in Figure 7. The loss began to converge after twenty epochs. The improved U-Net, U-Net++, ResU-Net, A-DenseU-Net, ResU-Net++, U-Net, Mask R-CNN, and DenseNet converged faster, while FCN, SegNet, and Deeplab V3 converged relatively slowly. After forty epoch trainings, the training loss was stable in a converging state. As a whole, the improved U-Net had a faster convergence speed and lower loss value, showing better performance.

### 3.6. Performance Comparison with Different Loss Functions

To solve the problem of extremely unbalanced image pixels, many improved loss functions have been proposed, and the effects of these loss functions on the symmetry of metallographic images have been studied. At present, few papers have done this work in the field of metallographic images. In this paper, five different loss functions were applied to the segmentation of metallographic images. The loss functions used in this paper include Cross entropy loss, Dice loss, Tversky loss, Tversky and Focal generalized loss, and Focal loss.

After the comparison of several groups of experiments, the best result of each loss function was selected for comparison, and the combination of Focal loss and Dice loss was finally selected as the loss function. When the parameter was set to *α* = 0.25, *β* = 2, the overall effect was the best, and the MIoU could reach 0.88. The adjustment factor reduces the impact of the loss of simple samples, reduces the weighting ratio of simple samples, and expands the acceptance range of low loss samples, so it is more favorable for the study of difficult samples. The final experimental results are shown in Table 4.

It can be seen that ACC of the six methods could reach more than 95%. Since the ACC evaluation indices are very similar, it is difficult to evaluate the performance of loss functions.

Compared with ACC, precision is more suitable for evaluating the performance of the metallographic image segmentation method because it avoids the interference of the large background. The experimental results show that the frame of this paper was reliable.

The experimental results also show that the loss function had a very important influence on the image segmentation effect. Among them, Focal loss and Dice loss were combined, and its parameter was set to 0.25 after many explorations, and the effect was best when it was set to 2. Among them, Loss, ACC, Precision, Recall, MIoU, Dice, and F1 were 0.102, 0.987, 0.931, 0.965, 0.88, 0.761, and 0.958, respectively. All the scores were relatively high, and the best results were achieved.

### 3.7. The Application of Grain Size Rating

The quantitative analysis of grain size is of great theoretical and practical significance to the study of materials. The American Society for Testing Material (ASTM) standard E112-12 recommends the planimetric and intercept methods for more accurate grain size measurements. The grain size *G* can be calculated as follows:(16)$$G=-3.321928\log_{10}N_{A}-2.954$$
(17)$$G=-6.643856\log_{10}\bar{l}-3.288$$

The planimetric method is shown in Equation (16), and the intercept method is shown in Equation (17). $N_{A}$ represents the total number of grains per unit area, and $\bar{l}$ represents the average of multiple cross-sections. The result of the planimetric method and intercept method is shown in Figure 8a and Figure 8b, respectively. It can be seen that the planimetric method and intercept method could produce the same level of precision.

In this paper, the planimetric method was used to calculate the grain size. To measure the grain area step:
Select a rectangular area and calculate its area, as shown in Figure 8a.Calculate the number of grains: $N=N^{\prime }+\frac{1}{2}N^{\prime \prime }$ ($N^{\prime }$ is the number of grains in the box, $N^{\prime \prime }$ is the number of grains across the box boundary, shown in Table 5). The result calculated by the formula method differs very little from the result manually calculated, and a high accuracy rate is achieved.The grain area is obtained by dividing the rectangular area by the number of grains.Calculate the number of grains in the unit area (mm^2^) and the grain size level, shown in Table 6.


Equation (16) gives the 95% confidence interval (95% *CI*) [51], and Equation (17) gives the relative accuracy percentage (*%RA*) [52], which can describe the accuracy of an automatic measurement.
(18)$$95\%CI=\pm \frac{ts}{\sqrt{n}}$$
(19)$$\%RA=\frac{95\%CI}{l}$$

As shown in Table 6, the difference between the numbers of grain measured manually and automatically was kept within 2. The accuracy was very high, and the error rate was no higher than 0.01. It can also be seen that the values of 95% *CI* and %*RA* were both less than 10. Therefore, the measured results were acceptable.

## 4. Conclusions and Future Work

An improved U-Net model with less training data was proposed to segment images. The model only trained with sixty images could produce satisfactory segmentation results, especially for fuzzy boundary metallographic images with a complex microstructure. The proposed method was compared with different image segmentation methods, including the Morphological method, the Canny operator, the Watershed method, FCN, SegNet, DeepLabV3, DenseNet, Mask R-CNN, U-Net, ResU-Net, A-DenseU-Net, ResU-Net++, and U-Net++. The comparison results showed that the improved U-Net was superior to the traditional metallographic image processing methods and other machine learning-based methods. It had higher scores on ACC, MIoU, Precision, and F1, with ACC of 0.97, MIoU of 0.752, Precision of 0.98, and F1 of 0.96.

## Figures and Tables

**Figure 1 materials-15-04417-f001:**
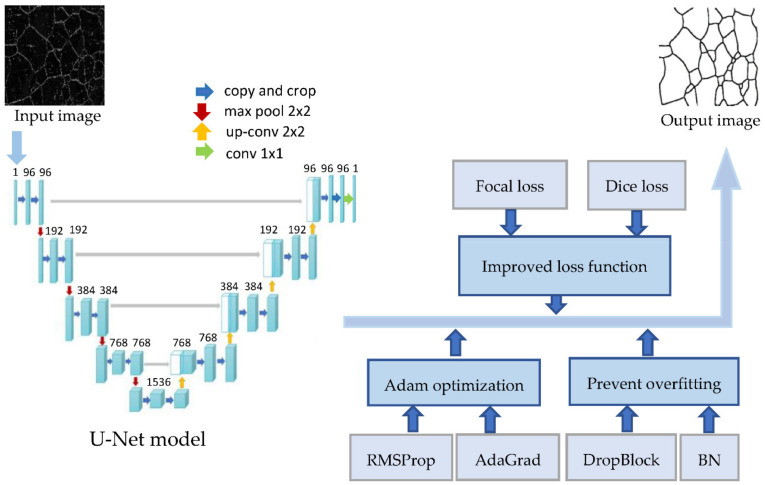
The structure of the improved U-Net segmentation model.

**Figure 2 materials-15-04417-f002:**
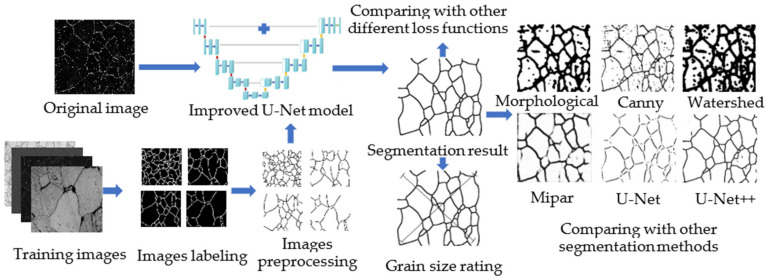
Specific work steps’ flow chart.

**Figure 3 materials-15-04417-f003:**
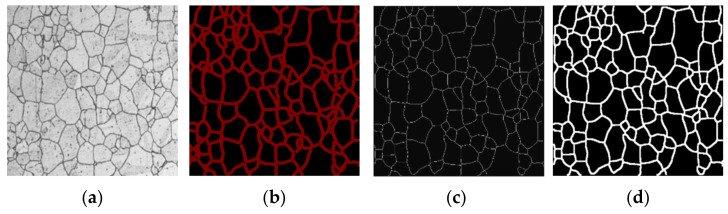
Process of image annotation. (**a**) Original image; (**b**) annotated image; (**c**) single-pixel skeleton image; (**d**) bolded image.

**Figure 4 materials-15-04417-f004:**
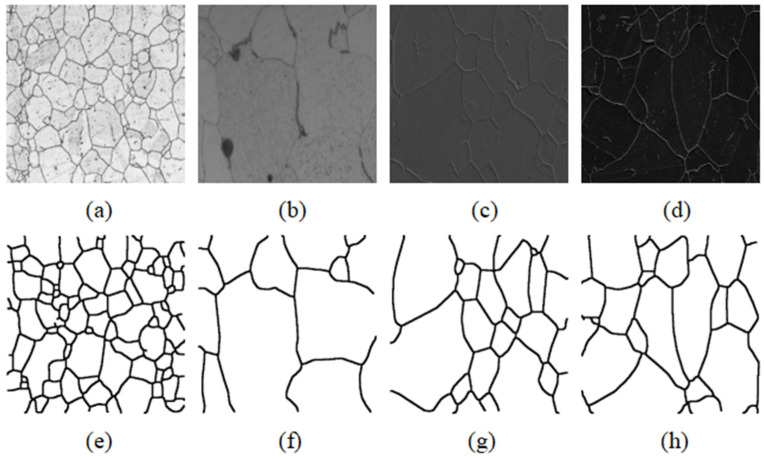
Examples of metallographic images. (**a**–**d**) Four metallographic images and (**e**–**h**) preprocessed images.

**Figure 5 materials-15-04417-f005:**
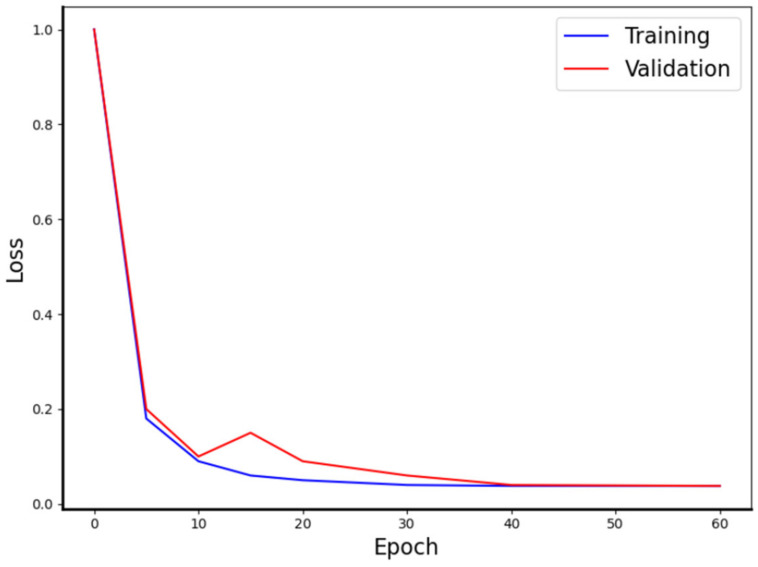
The training and validation loss of the model’s training.

**Figure 6 materials-15-04417-f006:**
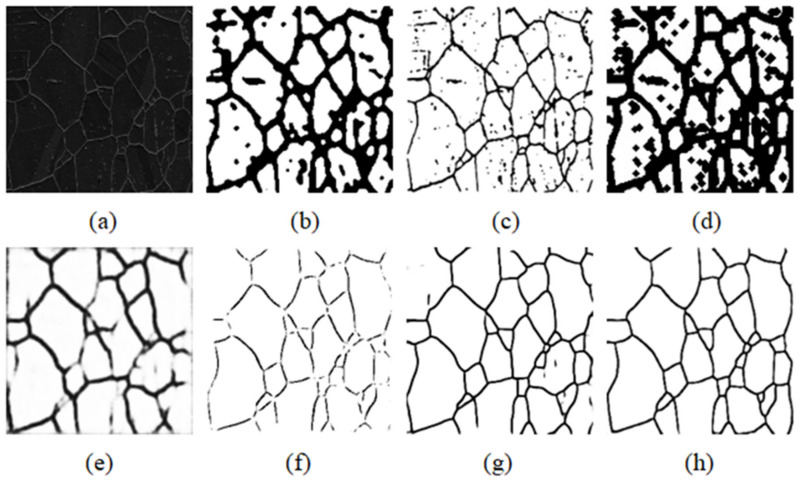
The segmentation result of different segmentation methods. (**a**) The original image; (**b**) Morphological methods; (**c**) Canny algorithm; (**d**) Watershed algorithms; (**e**) Mipar; (**f**) U-Net; (**g**) U-Net++; (**h**) improved U-Net.

**Figure 7 materials-15-04417-f007:**
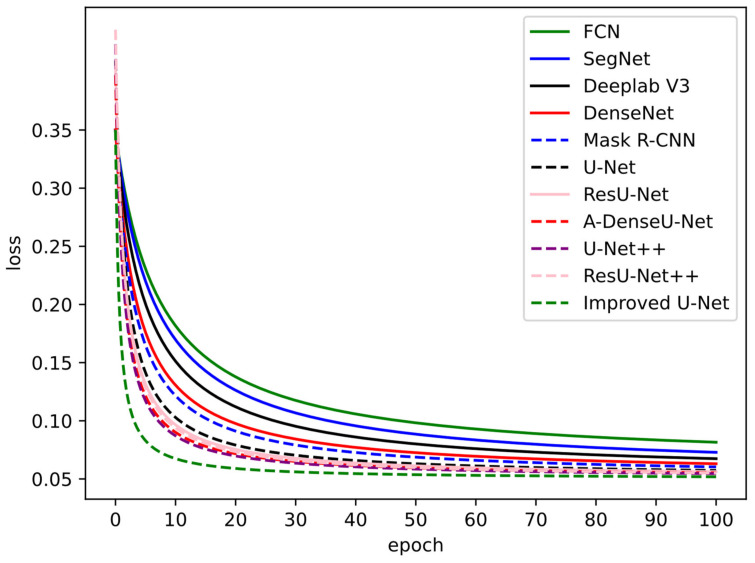
Training losses of different segmentation methods.

**Figure 8 materials-15-04417-f008:**
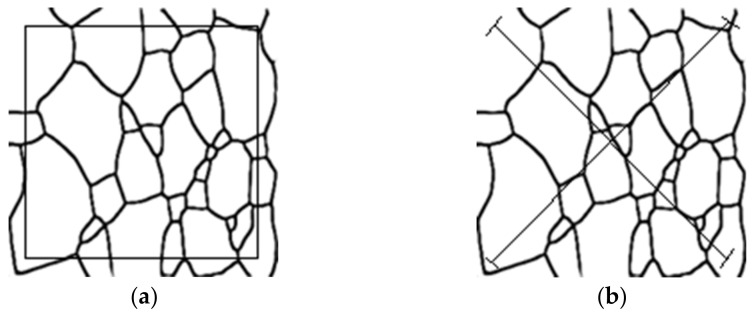
Grain size measurement results according to ASTM E112-12. (**a**) The planimetric method; (**b**) the intercept method.

**Table 1 materials-15-04417-t001:** The meanings of TP, FP, TN, and FN.

Terminology	Full Name	Meaning
TP	True positive	The number of positive examples predicted correctly.
FP	False positive	The number of positive example prediction errors.
TN	True negative	The number of negative examples predicted correctly.
FN	False negative	The number of negative example prediction errors.

**Table 2 materials-15-04417-t002:** Overall performance for different optimizers.

Optimizer	Learning Rate	ACC	Dice	MIoU	Precision
Adam	0.015	0.965	0.878	0.894	0.981
SGD	0.015	0.941	0.783	0.851	0.950
Adadelta	0.015	0.912	0.790	0.795	0.891
AdaGrad	0.015	0.934	0.817	0.610	0.887

**Table 3 materials-15-04417-t003:** The metrics’ results of different segmentation methods.

Methods	ACC	Dice	MIoU	Precision	Recall	F1
FCN	0.654	0.678	0.341	0.85	0.876	0.86
SegNet	0.679	0.783	0.451	0.95	0.853	0.898
Deeplab V3	0.91	0.790	0.595	0.734	0.983	0.840
DenseNet	0.89	0.917	0.610	0.657	0.942	0.774
Mask R-CNN	0.87	0.937	0.556	0.894	0.871	0.882
U-Net	0.89	0.910	0.700	0.904	0.880	0.892
ResU-Net	0.905	0.912	0.621	0.864	0.88	0.872
A-DenseU-Net	0.843	0.916	0.684	0.92	0.855	0.886
ResU-Net++	0.919	0.927	0.741	0.89	0.803	0.844
U-Net++	0.96	0.915	0.718	0.957	0.89	0.922
Improved U-Net	0.97	0.93	0.752	0.98	0.940	0.960

**Table 4 materials-15-04417-t004:** The performance of different loss functions on our model.

Loss Function	Loss	Acc	Precision	Recall	MIoU	Dice	F1
Cross entropy	0.134	0.967	0.650	0.558	0.335	0.483	0.600
Focal loss	0.631	0.973	0.859	0.122	0.013	0.403	0.210
Dice loss	0.654	0.952	0.832	0.807	0.342	0.786	0.819
Tversky	0.112	0.967	0.783	0.783	0.632	0.431	0.783
Focal and Dice	0.102	0.987	0.951	0.965	0.880	0.761	0.958
Tversky and focal Generalized	0.132	0.968	0.653	0.573	0.600	0.663	0.610

**Table 5 materials-15-04417-t005:** Typical metallographic grain number statistics in SGMD.

Image ID	Human	Automatic Calculation	Error Number	Error Rate
1	173	174	+1	0.008
2	90	90	0	0
3	110	114	+4	0.036
4	69	68	−1	0.014
5	200	200	0	0
6	151	152	+1	0.007
7	160	159	−1	0.006
8	171	170	0	0
9	300	300	0	0
10	190	188	−2	0.1
Average				0.017

**Table 6 materials-15-04417-t006:** Typical grain size level statistics in SGMD.

Image ID	G (ASTM)	±95% CI	%*RA*
1	6.90	0.163	5.41
2	5.70	0.266	8.32
3	7.36	0.151	5.02
4	7.89	0.201	6.04
5	7.95	0.321	10.1
6	7.90	0.131	4.51
7	6.68	0.163	5.41
8	6.73	0.204	6.21
9	6.92	0.210	6.30
10	7.50	0.109	3.01
Average	7.053	0.1919	6.1

## Data Availability

Access to download the code and data can be found on Dropbox: https://bit.ly/3zOaKtX; accessed on 18 June 2022.

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
