# Peer review of "An Improved U-Net Image Segmentation Method and Its Application for Metallic Grain Size Statistics"

_materials, 2022, doi:10.3390/ma15134417_

Round 1

Reviewer 1 Report

The manuscript is about an improved U-Net image segmentation method and its application for metallic grain size statistics. The proposed machine learning-based methods such as FCN, SegNet, DeepLabV3  need a large dataset to train the model. Improved U-Net model was applied in the present study for image segmentation of complex images with only a small training set. The manuscript has an important success rate and it is done with using small dataset.

However, the recommended comments below are important to make the manuscript suitable for publication.

1) In abstract;

The introduction is given about image segmentation and image processing methods. The method used in the study is mentioned. It is indicated that the experiments show significant advantage and others. The presentation is good until here.

However, it is important to present some numerical results.

For example, it is necessary to present the error rate or success rate.

2) For Introduction;

In Figure 1, there are blurred words. It is hard to read. Please, increase the quality and font sizes of words.

3) For Section 2;

How many training images are tested by the present method? Please, indicate in method section. Does the used method give successful results for all training images? If it is, please add the information in the paper.

4) For section 3;

It is necessary that the results must be discussed with more studies.

5) Title of tables 5 and 6 are different. However, the data is same. What is the difference of tables 5 and 6?

6) Conclusion section needs expanding and supporting with numerical results.

Author Response

Thank you very much for your suggestions. We have made corresponding revisions and supplements according to your suggestions. Please refer to the attachment for specific modifications.

Reviewer 2 Report

I have found the manuscript of interest. It worth publishing with some minor revisions which could be found on the attached pdf file.

The Introduction contains an up-to-date review of achievements in this field, being even unusually large (> 1100 words), close to a mini-overview, but I totally agree with its content. For this reason, excepting some small details, I haven’t commented this fact.

The next section: Improved U-Net, presented more detailed on how to better evidence the grain boundaries of a dark metallographic image. Here I have made a comment: really there are no other possibility to process the original image by adjusting the contrast, intensity and brightness ?

The results the authors provide in  Experiment results and discussion as well as in Conclusion and future work proves, in my opinion, that the improved U-Net model with less training data seems to give good results for normal size images. In fact, the AI method to be validated it should be made known, and then, let the other user to appreciate it at its true value.

For this reason I proposed the Manuscript to be accepted with minor revisions.

As I am familiarized with thin sections image in petrology, I think this procedure would be of interest if such studies would be used in petrology too.

Author Response

(The authors gave the same response as above.)

Reviewer 3 Report

Dear authors,

thank you for your well-written manuscript. You present a scientifically sound work which is of great interest for the community. Your introduction gives all the information which readers need even if they are not too much into image analyses / CNN.

I really liked your comparison with traditional image segmentation methods and more advanced methods and that you finally evaluate with ATSM standards against human manual analysis.

There is only one point I need to criticize: Since this is will be an OpenAccess publication, it will only complete if your data is published as well. This includes your CNN implementation or your trained model. Only this way, research on this important topic can go on.

You may also consider to publish your code / model with a seperate DOI, so that it can be cited.

Author Response

Thank you very much for your suggestion and your support of our work. We fully understand the necessity for open source code, which plays an important role in the continuation of our work. We promise that if the paper is accepted, we will publish our code and data as soon as possible.

Reviewer 4 Report

This manuscript is well organized and scientifically sound. I have a few comments for the authors:

1. Abstract. In line 14, the word "proposed" doesn't sound right. That should be "aforementioned".

2. In the first paragraph of the introduction section, references for each method have to be given. 

3. Avoid using "we" in the scientific journal paper.

4. Better resolution is required for Figure 1 and Figure 2.

5. The terms used in Equations (1)-(4) are not properly explained. For instance, what do the terms "N", "pi", "yi", "a", "TP", "FP", "A", "b", etc. stand for?

6. In line 169, what do the terms alpha and beta mean? 

7. Section 2.1 needs to be improved. Need to show more clearly what has been improved over the original loss function using equations.

8. The conclusion section is not solid. Emphasize major findings and do not include future work.

Author Response

(The authors gave the same response as above.)
